# FDA's PCCP: Opportunities and Gaps

No Author Given

No Institute Given

**Abstract.** The dynamic nature of medical imaging data poses a significant regulatory challenge for AI-based Software as a Medical Device (SaMD), as it requires constant adaptation. Traditionally, each of these modifications would require the SaMD to go through the complete approval process again, limiting the real-world deployment of AI-assisted SaMDs. The U.S. FDA's Predetermined Change Control Plan for Artificial Intelligence-Enabled Device Software Functions (PCCP) aims to bridge this gap. It allows, under certain conditions, a significantly simplified approval process for updated AI-enabled SaMDs. In this work, we discuss the great potential that this brings for the dynamic reality of medical imaging, but also explain three potential gaps in the current regulation. These concern an "evaluation gap" that poses a potential loophole from modified test data, an "intended use gap" that limits flexibility for unexpected and time-critical events such as a pandemic, and the "foundation model gap" that limits the applicability of this emerging technology. For each of those gaps, we present a solution to fully leverage the potential of PCCP as a regulatory framework enabling technologies that address the dynamic reality of medical imaging.

**Keywords:** PCCP · Continual Learning · Medical Imaging

## 1 Introduction

The demand for deploying Artificial Intelligence (AI) models for medical imaging is increasing, but the dynamic reality of medical image data poses a significant challenge [6], [14], [18], [10]. Existing technical solutions such as Continual Learning [9] allow to effectively train data that is constantly changing. However, *real-world deployment is hindered by the current state of regulations* that so far does not acknowledge this potential, as most authorities *require for already certified models to pass through the whole certification process again upon receiving training on additional data*, often making it impractical due to the long process that entails with an average duration of 338 days for a medium risk device [5] and high costs.

The U.S. Food & Drug Administration (FDA) has recently published the final version for the Marketing Submission Recommendations for a Predetermined Change Control Plan for Artificial Intelligence-Enabled Device Software Functions (PCCP) [3] that serves as a framework for a more dynamic certification process. It explains the process of only reviewing the changes to the model that

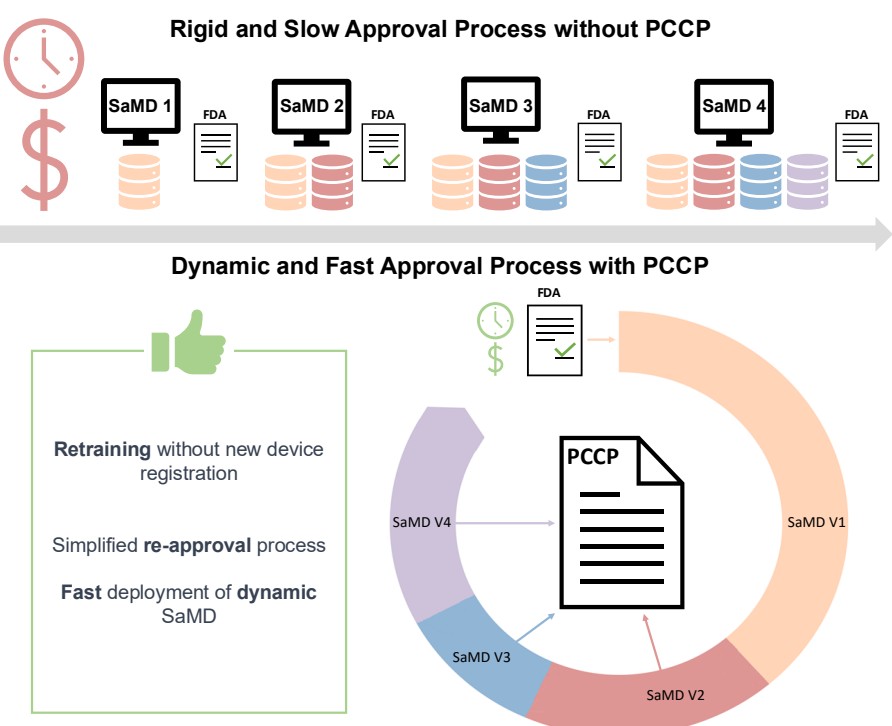

**Fig. 1.** *Opportunities of PCCP:* The traditional approval process for SaMDs (top) is slow and inflexible, as it requires a new marketing submission for each model update. This does not align with the dynamic reality of medical imaging, where training data constantly changes. PCCP improves the efficiency of the approval process in these dynamic scenarios.

were previously announced in the marketing submission, instead of requiring it to pass through the whole certification process again.

*This is a significant step towards regulatory-compliant deployment of continual learning models that are robust to the dynamic reality of medical imaging.* Figure 1 gives an overview of the improvements of PCCP compared to the traditional regulatory process.

However, there are also some potential issues in the current state of PCCP that we point out in this work, such as:

1. *Evaluation Gap:* This could potentially allow getting an update of a SaMD approved despite performance degradation on the test set that was used on the initial marketing submission.
2. *Intended Use Gap:* Limitations in flexibility that hinder the potential of PCCP for time-critical adaptations, such as in the case of a pandemic.

3. *Foundation Model Gap:* Due to the strict definition of the intended use, PCCP cannot be applied for Foundation Models, even though these are an important step to improve robustness of models on dynamic medical images.

Table 1 provides an overview of these issues and possible solutions to them. To the best of our knowledge, no existing publications are addressing these technical aspects of PCCP. Therefore, in the following, we provide details on these issues in the latest final version of PCCP as it was issued on December 4, 2024.

Moreover, we highlight potential changes that would mitigate these issues and allow to fully leverage the potential of PCCP for enabling an efficient approval process for SaMD that pays justice to the dynamic reality of medical imaging.

| Gap | Example | Solution |
|---|---|---|
| **Evalutation Gap** | Potential approval of model despite performance degradation on the earlier test set. | During marketing submission, define a performance goal for the initial test set |
| **Intended Use Gap** | Narrow intended use in urgent cases like pandemics with unknown diseases at marketing submission | Use alternative definition of intended use which defines manifestations of conditions and modalities |
| **Foundation Model Gap** | Intended use cannot be clearly defined for Foundation Models | Define performance goals for each initially defined use case and make sure to fulfill them during the evaluation of the PCCP |

**Table 1.** Overview of the issues of PCCP and their corresponding suggested solutions

## 2 Background

The Predetermined Change Control Plan (PCCP), is a novel regulatory framework, issued by the FDA on December 4, 2024, that allows AI-based SaMDs to be updated after the initial marketing submission unlike these other existing regulations. For the approval of such updates, a PCCP consists of the following three main components:

- *Description of Modifications*: Describes the future updates, such as retraining on additional data or other improvements.
- *Modification Protocol*: Defines how these changes stated in the description of modifications should be implemented safely. This includes information such as training data details, learning methods and performance benchmark methods.
- *Impact Assessment*: Assesses how each of the proposed modifications and the corresponding implementations might affect the safety and performance by providing components such as a risk analysis, clinical performance assessment, mitigation strategies, monitoring plans and rollback strategies.

The PCCP poses great potential to improve the regulatory process for SaMDs that require constant updates during their life cycle such as models that will be trained on additional data to adapt to the dynamic reality of medical imaging that might include population or acquisition shifts or new tasks. This is an important improvement compared to the traditional process that would require a new marketing submission for all changes, potentially taking too long for time-critical applications and binding significant resources. Existing regulations from other countries' regulatory bodies for SaMDs such as those from the European Union [4], China [1] and India [2] do not allow such simplified approval processes for updated medical devices, highlighting the novelty of the FDA's approach with PCCP.

## 3   The Evaluation Gap

**Issue:**  As mentioned before, PCCP poses great potential to enable a significantly simplified approval process of updated SaMDs that are trained in a Continual Learning process. For example, if new training data becomes available and the manufacturer is training the existing model on these additional training data, this new version of the model would have to be tested on a separate test set to confirm that the performance can be reliably achieved. Naturally, as part of the Continual Learning process, the test set would change as well compared to the initial training. However, *PCCP does not clearly define, what kind of changes are restricted for the test set*. Specifically, in section VII.B (1) (Data management practices) on page 25 it only requires providing an overview of the methods for collecting, organizing, storing, retaining and controlling the new data. A *manipulated, updated test set can potentially help to pretend a better performance of the model than it can actually provide*: For example, let us imagine a company that marks an approved SaMD capable of classifying pulmonary embolisms. This company wants to get an updated version of the model approved using PCCP. Section VII B (3) (Performance Evaluation) of PCCP on page 27 requires "AI model testing protocols comparing the newly modified device to both the original device (the version of the device without any modifications implemented) and the last modified version of the device". As visualized in Figure 2, this potentially allows changes to the SaMD to get approved that actually lead to a performance degradation on the initial test set, potentially compromising the safety of patients.

 **Possible Improvements:**  To mitigate this risk and avoid this loophole, we suggest adding the requirement for an *additional mandatory evaluation on the initial test dataset and all following datasets from previous already approved updates*. It should be required to retain a minimum performance that needs to be defined in the marketing submission for any future updates. This allows to tolerate a certain limited performance degradation as it is normal in Continual Learning while ensuring to stay in a margin that was approved by the FDA. Specifically, we would add in Section VII B (3) (Performance Evaluation) on

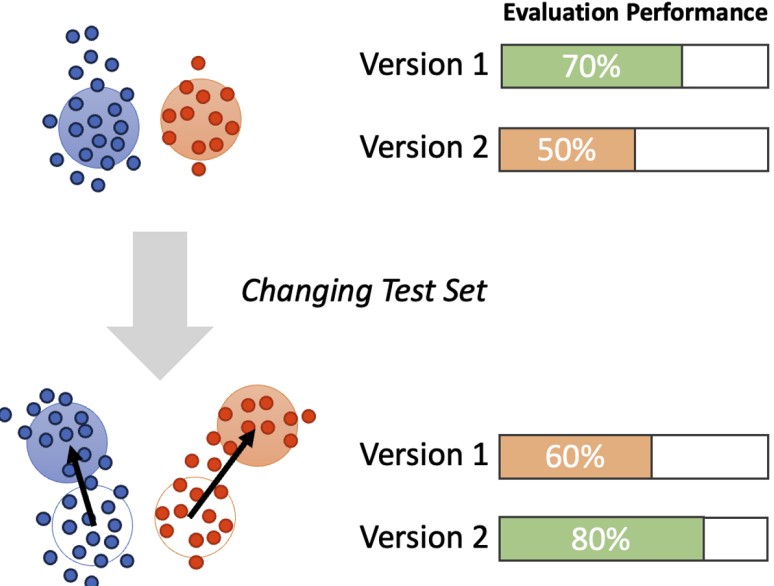

**Fig. 2.** *The evaluation gap:* By changing the distribution of the test data in a way that improves performance of the updated model on it, the manufacturer can create the impression that the update to the SaMD improved the performance while it could actually potentially still degrade on other test set distributions such as the distribution from the initial marketing submission.

page 27 of PCCP that the performance evaluation not only has to be conducted on the previous and new version of the product, but also on the previous and the potential new dataset. If the new version of the model does not retain the performance margin stated in the marketing submission on all datasets that it was previously evaluated on, its approval should be denied. This small change to the performance evaluation section of PCCP allows improving patient safety compared to the current PCCP, while still fully leveraging the potential of the significantly simplified regulatory approval process for continual model updates.

## 4   Intended Use Gap

**Issue:** Continual Learning approaches that allow to adapt models to new data distributions and new tasks, have great potential to quickly and reliably adapt existing AI models to changing environments and tasks, such as induced by the occurrence of a pandemic. For example, we assume to have a SaMD for classification of pulmonary embolisms (PE) and lung tumors captured in a CT successfully registered at the FDA. When time-critical scenarios like a pandemic, such as COVID-19 occurs where doctors face the challenge of distinguishing between Ground Glass Opacities (GGO) and COVID-19 cases that shares the manifes-

tation with these previously covered diseases, this device could potentially be a promising foundation for a new revision of the device that was continually trained on the new train set with this new task and the pandemic data distribution. Specifically, the existing model could be trained in a Continual Learning setup on a third task (COVID-19) additionally to the already approved classification of lung tumor and PE.

This would technically require minimal effort, but unfortunately PCCP Section VI C on page 22 limits such a modification as it states, "Modifications included in a PCCP must maintain the device within the device's intended use". This "intended use" is defined in sections 515C(a)(2) and 515C(b)(2) of the FD&C Act. Adding a new task such as COVID-19 detection would violate this requirement and also would be unknown during the initial approval process that happened before this new disease occurred for the first time. *Therefore, it would still be required to go through the whole approval process again and register this extended version of the model as a new SaMD.* This limitation is *problematic as a situation such as a pandemic requires a quick response* and the re-approval process could potentially take too long to help during the critical phase of the pandemic. Figure 3 visualizes this problem.

**Possible Improvements:**   Even though it is not easy to have a full proof regulation for this kind of complex problem, that requires to balance the goal of preserving performance on old data while also allowing improved performance on new data or covering new tasks/diseases. A possible solution would be to *add an exception for PCCP regarding the definition of the intended use beyond the currently used definitions that are defined in sections 515C(a)(2) and 515C(b)(2) of the FD&C Act*: It should additionally allow defining a list of diseases and their manifestations combined with a list of modalities during the marketing submission.

To improve flexibility, it should be possible, as part of PCCP, to add new diseases that share manifestations with the initially submitted list. For safety, the modified definition of the intended use should still require keeping the initial data modality or task type (classification, segmentation etc.). In the previously described pandemic scenario, this would allow benefiting from the simplicity of a PCCP, as the modality (lung CTs), type of task (classification) and manifestation would remain the same as in the initially approved marketing submission of the SaMD. As only the training data changed by adding a new class of COVID-19, with this suggested broader definition of the intended use, the updated SaMD would now be eligible for a PCCP instead of going through a new complete approval process. This would allow to quickly react to new changes such as a pandemic while preserving patient safety.

## 5   Foundation Model Gap

**Issue:** As stated in section V (Policy for Predetermined Change Control Plans) on page 10, the FDA demands precise specifications for the intended use of a

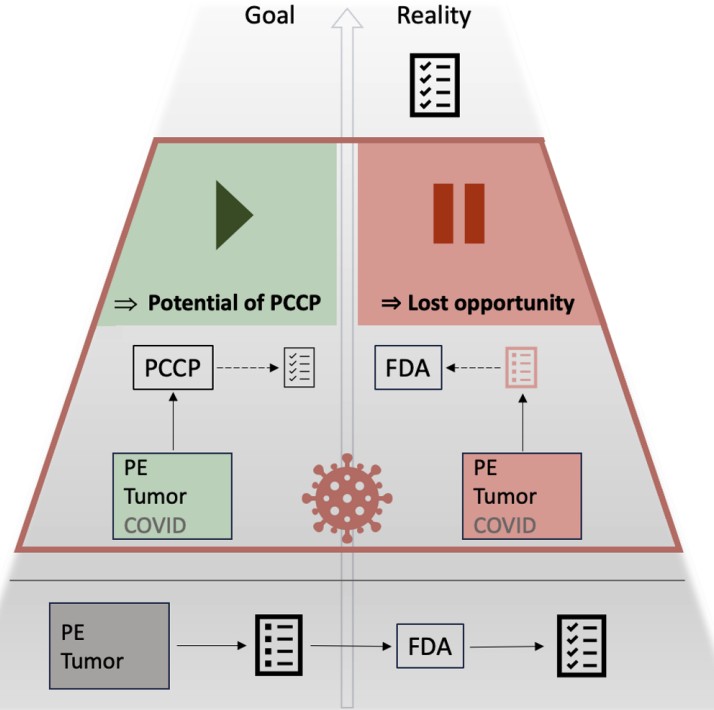

**Fig. 3.** *The intended use gap:* In time-critical events such as a pandemic, the PCCP could potentially accelerate the approval process by training an existing and similar FDA-approved SaMD on the new pandemic datasets. However, the current version of PCCP strictly requires preserving the device's intended use, which currently blocks PCCP to being applied in this scenario. This is a lost opportunity, as a swift approval is required during pandemics.

device and generally requires a new marketing submission in case of any changes of it ("Manufacturers should evaluate the impact of modifications to their devices and must generally submit a marketing submission when device modifications affect the intended use of the device or [...]"). This makes sense for traditional applications and models but hinders the use of modern, emerging concepts such as Foundation Models. The recent fast growth of medical Foundation Models [17,12,13,7] shows the demand and importance of this technology for the field. They bring significant potential for SaMDs as they enable advanced capabilities like image interpretation and predictive analytics. Their ability to learn from vast, diverse datasets helps improve diagnostic accuracy and support clinical decision-making across a wide range of conditions. Moreover, there are already applications to deploy Foundation Models such as the Segment Anything Model (SAM) [8], Medical SAM [16] and to enable domain-incremental Continual Learning [11] with a strong potential for task-incremental learning [15].

Unfortunately, due to this wide range of conditions, modalities and data, the current definition of the intended use cannot be clearly defined in a marketing submission for a medical Foundation Model. However, the current version of the PCCP requires a clear definition of a device's intended use during marketing submission, *making it impractical to get Foundation Models approved*. This means that the *significant potential of this important trend in AI-assisted SaMDs cannot be leveraged in FDA-approved real-world applications, hindering progress in the field*.

**Possible Improvements:**   To mitigate this limitation, we suggest broadening the definition of the intended use of SaMDs: If a group of modalities and tasks could be mentioned in the initial marketing submission combined with the information that the SaMD is based on a Foundation Model, it should be possible to evaluate the performance and safety on exactly these data and tasks. Even though, the evaluation would be complex, it would enable the deployment of limited Foundation Models, balancing the demand for safety and flexibility. If additional modalities or tasks should be added, a *PCCP must allow adding these under the condition that the updated SaMD fulfills the performance range on all the tasks from the initial evaluation as defined in the initial marketing submission*. This change would enable the safe and regulated deployment of these emerging techniques in the clinical reality.

## 6    Conclusion

We gave a technical overview of the PCCP framework and explained what potential it brings for training in dynamic environments with dynamically changing medical imaging data. We explained three potential issues that we see with the current version of PCCP for these dynamic applications.

The first issue concerns a potential manipulation to the advantage of the manufacturer by *modifying the test set* for the evaluation of the updated SaMD. We suggested an additional evaluation on old test data as a potential improvement to close this loophole while still reporting evaluations on updated test sets.

The second stated issue is that the definition of the "intended use" of the SaMD is *too strict to swiftly react to time-critical and unexpected events such as pandemics*. We suggested broadening the definition of the "intended use" for PCCP in a way that allows to leverage PCCP in such time-critical situations while still ensuring patients' safety.

The third and last stated issue is that the current version of PCCP is *conflicting with the important trend in AI on medical imaging that is Foundation Models*, as their intended use cannot be defined precisely enough as required by PCCP. Our suggested change to the definition of the intended use mitigates this problem.

All the proposed improvements described in this work make it possible to fully leverage the great potential that PCCP offers as a regulatory framework to support these technologies that address the dynamic reality of medical imaging.

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
