# OpenReview forum: "FDA’s PCCP: Opportunities and Gaps"
_MICCAI.org/2025/Workshop/BRIDGE — BRIDGE 2025 Poster_

### Official Review · Reviewer_2rJf · 2025-07-22
**FDA’s PCCP: Opportunities and Gaps**

**Rating:** 8
**Confidence:** 5

**Review:**

Summary
This paper presents a solution to  leverage the potential of the Predetermined Change Control Plan (PCCP) as a regulatory framework for enabling technologies that address the dynamic reality of medical imaging. The analysis is timely and addresses a highly relevant regulatory challenge in medical AI deployment. IT addresses an important and timely topic with good use of practical examples.

Strengths:
1. The identified gaps and proposed solutions are justified and have clear real-world relevance.
2. The discussion is highly pertinent to the BRIDGE workshop objectives.
3. The PCCP framework is relatively new and would benefit from further refinement, making the paper valuable.
4. The paper effectively uses concrete scenarios (e.g., COVID-19 pandemic, pulmonary embolism detection) to illustrate the practical implications of the identified gaps.
5. These real-world examples strengthen the paper's arguments and demonstrate practical relevance

However, the proposed solutions lack sufficient consideration of implementation difficulties and potential unintended consequences of the suggested regulatory changes. For example, updating intended use without full clinical evaluation presents serious risks: while this approach offers practical flexibility, changes to intended use can alter the device's risk profile and introduce new, unforeseen risks.

The paper would benefit from better balancing practical needs with patient safety considerations, as the intended use fundamentally defines the risk profile for medical devices. One way to do this is to discuss how each suggested improvement can guarantee patient safety.

Reference numbering appears to contain errors and should be corrected, and there are some writing errors.

---

### Official Review · Reviewer_DTSR · 2025-07-23
**Good, timely paper that can start interesting discussions**

**Rating:** 8
**Confidence:** 4

**Review:**

1. Summary of the Paper

This paper discusses the recently introduced FDA PCCP pathway. The authors identify three gaps that would be good to address in further versions of the guidelines.

2. Strengths

- This paper is timely and addresses a regulatory area that many AI companies are currently looking into and have questions about.
- The paper is well written and well structured and the graphics are very helpful to understand the problems that the authors are addressing.
- This paper can start the necessary discussion to further advance the PCCP regulatory pathway.

3. Limitations or Areas for Improvement

- Throughout the paper the term 'continual learning' is used. I would suggest using a slightly different term (e.g., offline continual learning) or maybe explain the term briefly.
Continual learning is often used to refer to a system that adapts on-the-fly, during deployment (also referred to as online continual learning). So in a medical setting, this would be a system that updates itself while it is deployed in a clinic, for example by using feedback from radiologists operating the system. This is not allowed under PCCP, to the best of my knowledge.

The graphic in figure 1 makes it clear that this is also not what the authors are referring to, but the term might cause confusion.

- Regarding the suggested mandatory evaluation in the original test set
This is an interesting point, but aren't there situations where it is fine to actually have a decrease in performance on the initial test set (e.g., if the old test set includes device manufacturers that are no longer used or if there are changes in the global patient population that would make the test set outdated?).

Fully agree that it would be good to have clear guidelines for this from the FDA.

Another minor suggestion here could be to use the term 'validation set' as I think the FDA usually works with 'training', 'tuning' and 'valdiation'.

- 'Foundation model gap'
This is an interesting point, but from my perspective, a foundation model is essentially just a pre-trained backbone that can be adjusted to new problems (in particular new tasks). I think here, the issue in the PCCP pathway is not so much the foundation model, but rather that new tasks (e.g., classification of new diseases, etc) are not allowed as it would change the intended use.

If the intended use is the same, I would expect the FDA to allow the addition of a foundation model to a PCCP, as it would not be much different from a change in backbone.

It might be good to make this more clear and identify how this gap differs from the second gap (intended use gap) as I think this is the root of the problem.

4. Relevance to BRIDGE Workshop Topics

- This paper is very relevant for the BRIDGE workshop. It address potential points of improvement in a regulatory pathway for AI/ML software.
- This paper can spark interesting discussions at the workshop.

---

### Official Review · Reviewer_1byk · 2025-07-25
**Significant misunderstandings of the PCCP guidance document and existing regulatory pathways**

**Rating:** 4
**Confidence:** 5

**Review:**

## Summary of the Paper
This position paper analyzes the FDA's Predetermined Change Control Plan (PCCP) framework for AI-enabled medical devices, identifying three purported critical gaps: an "evaluation gap," an "intended use gap," and a "foundation model gap." However, the paper demonstrates significant misunderstandings of the PCCP guidance document and existing regulatory pathways, undermining the validity of the identified gaps and proposed solutions.

## Strengths
- Addresses Timely Regulatory Topic: The paper focuses on the recently finalized PCCP framework (December 2024), which is highly relevant to current AI/ML medical device development and deployment challenges.
- Clear Writing for Mixed Audience: The paper effectively communicates complex regulatory concepts in accessible language suitable for the BRIDGE workshop's multidisciplinary audience.
- Structured Analysis Approach: The authors provide a systematic examination of perceived regulatory gaps with proposed solutions, demonstrating an organized analytical framework.

## Limitations or Areas for Improvement
- Fundamental Misunderstanding of PCCP Requirements: The authors misinterpret key PCCP provisions. For the "evaluation gap," they claim PCCP doesn't define test set restrictions, but the guidance explicitly addresses data management practices requiring test data to be "representative of the current patient population and standard of care."
- Ignores Existing Regulatory Pathways: The "intended use gap" argument overlooks established emergency authorization processes (e.g., Emergency Use Authorization) that already provide rapid approval mechanisms for pandemic responses, making their proposed solution redundant.
- Unclear Foundation Model Analysis: The authors fail to articulate how foundation models inherently change a device's intended use, conflating the underlying AI technology with the clinical application and intended use statement.
- Lack of Regulatory Expertise: The analysis suggests insufficient familiarity with FDA regulatory processes and existing guidance documents, undermining the credibility of the proposed regulatory modifications.
- No Empirical Validation: The paper provides only hypothetical scenarios without real-world evidence that these "gaps" have created actual regulatory barriers or safety concerns.

## Relevance to BRIDGE Workshop Topics
- Moderate Relevance - While the paper addresses BRIDGE themes of regulatory frameworks and AI deployment challenges, the fundamental misunderstandings of existing regulations significantly diminish its contribution to advancing regulatory science discourse.

---

### Decision · Program_Chairs · 2025-07-25

**Decision:**

Accept (Poster)

**Comment:**

Dear Authors,

We are pleased to inform you that your paper has been accepted for the BRIDGE Workshop.
Your paper provides a valuable starting point for discussing this important topic and offers an opportunity for workshop participants to engage in discussing PCCP-related challenges. Your submission was reviewed by three scientists from regulatory, academic, and industry backgrounds to offer different perspectives that align with the workshop's interdisciplinary objectives. The reviewers' comments are provided below.

Requirements for your final camera-ready submission (due July 30):
* Incorporate reviewer comments and suggestions where appropriate throughout your paper. At minimum, add a discussion section that acknowledges and responds to the key points raised by reviewers
* Ensure your final draft follows standard MICCAI conference and Springer formats and guidelines
* Please submit your camera-ready source file, and any supplementary material you might have.


We look forward to your presentation and the discussions it will generate at the workshop.

Best regards,
BRIDGE Workshop Organizers